# Prevalence of Cardiovascular Risk Factors in Middle-Aged Lithuanian Men Based on Body Mass Index and Waist Circumference Group Results from the 2006–2016 Lithuanian High Cardiovascular Risk Prevention Program

**DOI:** 10.3390/medicina58121718

**Published:** 2022-11-24

**Authors:** Egidija Rinkūnienė, Emilija Petrulionytė, Vilma Dženkevičiūtė, Žaneta Petrulionienė, Augustė Senulytė, Roma Puronaitė, Aleksandras Laucevičius

**Affiliations:** Faculty of Medicine, Vilnius University, 03101 Vilnius, Lithuania

**Keywords:** risk factors, prevalence, body mass index, waist circumference, ABSI, obesity

## Abstract

*Background and aims*: This study aimed to estimate the prevalence of cardiovascular risk factors in middle-aged Lithuanian men categorized according to body mass index and waist circumference results. *Methods and results*: The data were from the Lithuanian High Cardiovascular Risk primary prevention program between 2009 and 2016. This community-based cross-sectional study comprised 38,412 men aged 40 to 54 years old. We compared the prevalence of arterial hypertension, dyslipidaemia, diabetes mellitus, smoking, and metabolic syndrome in body mass index (BMI) and waist circumference (WC) groups. Regarding the allometric anthropometrics for WC, A Body Shape Indices (ABSIs) were analyzed with respect to mortality risk and smoking status. The most prevalent risk factor in men was dyslipidaemia, followed by arterial hypertension and smoking (86.96%, 47.94%, and 40.52%, respectively). All risk factors except for smoking were more prevalent in men with overweight or obesity as measured by BMI compared to men with normal weight. Similarly, smoking was the only cardiovascular risk factor that was more prevalent among subjects with normal WC compared to those with increased WC or abdominal obesity. Elevated ABSI, which is associated with higher mortality risk, was more prevalent in smokers. *Conclusion*: The most prevalent cardiovascular risk factor among middle-aged Lithuanian men was dyslipidaemia, with a surprisingly high prevalence in all BMI and WC groups. Smoking was the only risk factor most prevalent in subjects with low or normal weight according to BMI. It was also more prevalent in the normal WC group compared to the increased WC or abdominal obesity groups, but ABSI values associated with higher mortality were more prevalent among smokers than non-smokers.

## 1. Introduction

Obesity is widely recognized as a pandemic public health problem. According to the World Health Organization (WHO), the rate of obesity has tripled since 1975, with 39% of adults being overweight and 13% being obese in 2016 [1]. Its prevalence is expected to peak among European and American men between 2030 and 2052 [2].

WHO defines obesity and overweight as abnormal or excessive fat accumulation that may impair health [3]. Excess adiposity greatly increases chronic disease morbidity and mortality, including cardiovascular disease (CVD) [4]. In high-income countries, over the past 60 years, the increase in life expectancy and decrease in premature mortality can be largely attributed to the decline in CVD mortality [5]. This has been achieved mainly through preventive measures—decreasing smoking and effective control of cholesterol levels and arterial blood pressure—as well as improved treatment due to the development of statins and thrombolytics and advances in treating acute coronary syndrome [6]. However, recent evidence suggests that the long-term decline in CVD and heart disease mortality may be stagnating, with the rates reversing in some populations. The rapid growth of obesity and other related risk factors are thought to be responsible for this trend [5]. Although adiposity measures, such as body mass index (BMI) or waist circumference (WC), are not the best predictors for CVD when compared with other cardiovascular risk factors, increased adiposity is strongly associated with the development of insulin resistance, diabetes mellitus (DM), dyslipidaemia, and hypertension, which are commonly used in many cardiovascular risk assessment tools [7]. Even though obesity is more prevalent in women, in Europe the potential gain in life expectancy by the elimination of obesity is higher in men [8]. In 2015, Lithuania had the widest gender gap in life expectancy in the European Union, with men having more than 10 years shorter life expectancy than women [9]. One of the many explanations for this phenomenon is the gender difference with respect to CVD and cardiovascular risk. Both the effects of androgens and the typical fat distribution pattern in males—accumulation of visceral fat—are associated with higher cardiovascular risk, which may explain why CVD typically presents much earlier in men [10].

BMI is the most widely used measure of obesity due to its simplicity and reproducibility [3,7]. However, it does not account for fat distribution. Fat accumulation in the trunk is associated with atherogenic dyslipidaemia, type 2 DM, hypertension, and inflammation, all of which increase the risk of CVD [11]. WC is arguably a more accurate indicator of visceral fat accumulation and of an adverse metabolic profile than BMI [7]. Nevertheless, the two indices are strongly correlated and therefore have similar associations with cardiovascular risk [12].

The purpose of this study was to determine the prevalence of arterial hypertension, dyslipidaemia, DM, smoking, and metabolic syndrome (MetS) in middle-aged Lithuanian men categorized by BMI and WC.

## 2. Material and Methods

The Lithuanian High Cardiovascular Risk Program (LitHiR) is a primary prevention program for middle-aged men and women designed to identify patients at high risk of cardiovascular disease and implement methods of primary prevention. LitHiR has been active since 2006. In 2016, 91.6% of primary healthcare institutions in Lithuania were taking part. Patients included in the program undergo a physical examination, assessment of risk profile and lifestyle (smoking, physical activity, dietary habits), anthropometric measurements, laboratory testing, 12-lead ECG, and have their SCORE index calculated. Lastly, based on the results of the investigation, the patients are divided into two groups: low–moderate risk and high risk [13].

Data for 38,412 men, aged 40 to 54, who participated in the LitHiR from 2006 to 2016, are presented in this report. Participants were divided into groups based on their BMIs and waist circumferences. BMI was calculated using the formula: $=\frac{\mathrm{weight}\left(\mathrm{kg}\right)}{{\mathrm{height}}^{2}\;\left(m^{2}\right)}$. Participants with BMIs lower than 18.5 kg/m^2^ were considered low-weight, a BMI of 18.5–24.99 kg/m^2^ was considered as normal weight, 25.0–29.99 kg/m^2^ as overweight, 30.0–34.99 kg/m^2^ as grade 1 obese, 35.0–39.99 kg/m^2^ as grade 2 obese, and 40 kg/m^2^ or above as grade 3 obese. Waist circumference (WC) was measured just above the iliac crest using a centimetre strip while the patient was standing and breathing normally. A WC of less than 90 cm was considered normal, 90–102 cm as increased, and more than 102 cm as abdominal obesity. To better understand the association between WC and smoking, we supplemented our analysis with calculations of A Body Shape Indices (ABSIs), which represent WCs adjusted for BMIs, and ABSI Z-scores, initially derived from NHANES, USA [14]. ABSI was calculated using the formula: $=\mathrm{WC}\times {\mathrm{weight}}^{-2/3}\times {\mathrm{height}}^{5/6}$. ABSI Z-score was calculated using the formula: $=\frac{ABSI-ABSI_{mean}}{ABSI_{SD}}$. Blood pressure was measured in a sitting position after at least 5 min of rest. The dominant arm was positioned at heart level and suitable cuffs were used. Arterial hypertension was determined when systolic blood pressure was ≥140 mmHg or the diagnosis of it was documented previously. Serum total cholesterol (TC), high-density lipoprotein cholesterol (HDL-C), and triglycerides were evaluated, and dyslipidaemia was diagnosed in accordance with the European Guidelines on CVD prevention, when any of the following was present: TC > 5 mmol/L, LDL-C > 3 mmol/L, HDL < 1.0 mmol/L, or TGs > 1.7 mmol/L. Genetic dyslipidaemias have not been taken into account in our analysis. Diabetes was determined from previous medical records. Metabolic syndrome was diagnosed in the presence of at least 3 out of 5 risk factors using the National Cholesterol Education Program III modified criteria (increased waist circumference > 102 cm for men, TC > 1.7 mmol/L, HDL-C < 1.03 mmol/L, systolic blood pressure ≥ 130 mmHg or diastolic blood pressure ≥ 85 mmHg or the patient is taking antihypertensives, fasting glucose ≥ 5.6 mmol/L). The study protocol was approved by the Vilnius Regional Biomedical Research Ethics Committee (no. 158200-15-816-329). There was no possibility of obtaining written informed consent from each patient, which was stated in our study protocol submitted to the Regional Biomedical Research Ethics Committee.

IBM SPSS 17.0 and Excel 16.66.1 software was used to perform the data analysis. Quantitative variables with normal distributions were presented as means ± standard deviations (SDs). *t*-tests or non-parametric Mann–Whitney tests were used to compare quantitative variables between two groups. Qualitative variables were presented using absolute frequencies (n) and percentages (%) of analyzed samples. Chi-squared and Fisher’s tests were used to compare the qualitative variables. Results were considered statistically significant when *p* was lower than 0.05. The normality of the distribution of continuous variables was tested using the Kolmogorov–Smirnov test. Although the data did not follow a normal distribution (*p* < 0.001), the variables were considered to be approximately close to a normal distribution due to the large sample size of the study, taking into account that parametric tests work well with large samples even when the distribution is non-normal [15]. Microsoft Word 2021 and Microsoft Excel 2021 were used to provide visual and graphical data.

## 3. Results

### 3.1. Sample Characteristics

Data for 38,412 men were included in the study. The mean age was 46.96 ± 4.39 years. Other characteristics of the study sample are presented in Table 1.

In the studied population, the most prevalent risk factor in men was dyslipidaemia, followed by arterial hypertension and smoking (86.96%, 47.94%, and 40.52%, respectively). According to BMI results, 28.36% percent of the men were obese, while a similar number (28.25%) were obese based on WCs (Figure 1).

### 3.2. Characteristics of The Study Sample Based on BMI

Of the men studied, 28.42% (*n* = 10,916) had a normal weight, 0.42% (*n* = 162) were low-weight, 42.74% (*n* = 16,417) were overweight, 20.92% (*n* = 8036) had grade 1 obesity, 5.72% (*n* = 2198) had grade 2 obesity, and 1.78% (*n* = 683) had grade 3 obesity (Figure 2). Evaluation of general characteristics showed that the only statistically non-significant difference between the normal-weight and overweight groups was heart rate (HR) (*p* = 0.982). When the overweight and grade 3 obese groups were compared, there was no difference found in total cholesterol (TC) or high-density lipoprotein cholesterol (HDL-C) between the groups (*p* = 0.499 and *p* = 0.198). There were also no significant differences in fasting glycemia, HDL-C, or SCORE indexes between the low-weight and normal-weight BMI groups (*p* = 0.763, *p* = 0.999, and *p* = 0.996) (Table 2).

#### Cardiovascular Risk Factors in BMI Groups

The most prevalent cardiovascular risk factor in all BMI groups was dyslipidaemia (66.05–93.45%). Comparisons between normal BMI and overweight groups as well as normal BMI and grade 3 obesity groups showed that arterial hypertension, dyslipidaemia, DM, abdominal obesity, and MetS were more common in overweight and grade III obesity groups (*p* < 0.001), while smoking was more common in the normal BMI group (*p* < 0.001). Smoking was also more common in the overweight group than in the grade 3 obesity group (*p* < 0.001 and *p* = 0.044). Overall, the highest frequency of smoking among the participants was observed in the low-weight group (61.11%). Comparison of the normal-weight and low-weight BMI groups showed no significant differences in the prevalences of cardiovascular risk factors, except for dyslipidaemia, which was more common in the normal BMI group (*p* = 0.001) (Figure 3).

### 3.3. Characteristics of the Study Sample Based on WC

Normal waist circumference was found in 33.80% of subjects (*n* = 12,983), increased waist circumference in 37.95% (*n* = 14,576), and obesity in 28.25% (*n* = 10,853) (Figure 4). The comparison between all three WC groups showed statistically significant differences in all the general characteristics, as shown in Table 3.

#### Prevalences of Cardiovascular Factors in WC Groups

Analysis of cardiovascular risk factor prevalences in the different WC groups showed similar trends to those observed for the BMI groups: the most prevalent risk factor in all groups was dyslipidaemia (80.37–92.51%). Men in the obesity group in comparison with those in the normal WC group were more likely to have arterial hypertension (29.72% (*n* = 3859) vs. 70.56% (*n* = 7658); *p* < 0.001), dyslipidaemia (80.37% (*n* = 10,434) vs. 92.51% (*n* = 10,040); *p* < 0.001), DM (6.55% (*n* = 850) vs. 19.98% (*n* = 2168); *p* < 0.001), and MetS (5.21% (*n* = 676) vs. 68.46% (*n* = 7430); *p* < 0.001), but smoking was more common in the normal waist circumference group (48.80% (*n* = 6336) vs. 35.04% (*n* = 3803); *p* < 0.001). Comparison between normal and increased WC groups showed the same trend: all risk factors were more common in the increased waist size group (*p* < 0.001), and smoking was more common in the normal waist size group (48.80% (*n* = 6336) vs. 37.22% (*n* = 5425); *p* < 0.001) (Figure 5).

### 3.4. ABSI and Smoking

Analysis of ABSI and ABSI Z-score mortality risks demonstrated that ABSI values associated with a higher risk of mortality were more prevalent in smokers compared to non-smokers (*p* < 0.001) (Figure 6).

## 4. Discussion

In our study, the most prevalent cardiovascular risk factor was dyslipidaemia. It had an unexpectedly high prevalence in all BMI and WC groups and was the only risk factor that significantly differed between the low-weight and normal-weight BMI groups. S. Kutkienė et al. demonstrated that men with dyslipidaemia were twice as likely to be obese as those without (30.3% vs. 16.1%, *p* < 0.001) [16]. The NHANES study reported similar findings: dyslipidaemia was more common in individuals with higher BMIs [17]. We found that blood lipid concentrations were increased in men with higher-than-normal BMIs and WCs (except for HDL-C). The findings of other, similar studies have shown that increased BMI and WC are associated with higher levels of LDL-C and TGs and decreased HDL-C [17,18,19].

The prevalences of hypertension, DM, and MetS were significantly higher in those with increased BMI and WC. Various studies have confirmed the association between increased BMI and WC and hypertension, DM, and MetS [17,20,21,22,23]. Out of these three risk factors, DM possibly has the strongest association with obesity, which increases the risk of type 2 DM even in the absence of other cardiovascular risk factors [24]. One study found that the risk of DM is increased three-fold with a BMI of 25–29.99 kg/m^2^ and 20-fold with a BMI of more than 30 kg/m^2^ [25]. The relationship between obesity and DM can be explained by structural and functional changes in adipose tissue. They lead to adipokine production by adipocytes and macrophages and later to insulin resistance [26].

Our findings show that smoking is significantly more prevalent in men who are underweight or of normal weight compared to those with overweight or obesity, according to BMI. Similarly, smoking was more common among subjects with normal WCs than those with increased WCs and abdominal obesity. To better understand this paradoxical relationship between WC and smoking, we calculated ABSIs and ABSI Z-scores [14]. Men who were smokers had a higher mortality risk as indicated by their ABSIs compared to non-smokers (*p* < 0.001). Previous studies have reported on the relationship between smoking and weight. Current smoking is generally associated with lower BMI than never smoking [27,28]. Some studies have reported the association of smoking with lower BMI and coinciding central obesity, but this association was observed to be less pronounced in men than in women [29,30,31]. Nicotine is a known appetite suppressor, which may explain why smoking is more prevalent among underweight and normal-weight subjects [32]. Furthermore, smoking cessation is associated with weight gain, which may contribute to higher BMIs among non-smokers [33,34]. Our study does not account for former smoking, which could be an interesting aspect to explore in relation to BMI and WC in the future. To our knowledge, there are no published studies reporting on the relationship between smoking and ABSI.

Due to its simplicity, BMI is the most widely used measure to evaluate body weight. However, when it comes to diagnosing obesity, BMI correlates poorly with percentage of body fat in males and does not account for its distribution [35]. Meta-analysis of 32 studies showed that commonly used BMI cut-offs to diagnose obesity have a pooled high specificity (0.9) but rather low sensitivity (0.5) [36]. Some studies found that WC might be a better predictor of cardiovascular risk factors and CVD in men [37,38]. Even though BMI is not always an accurate predictor of obesity, BMI is associated with atherosclerotic CVD and mortality risk [7,39]. However, in some studies, when such correlations are found, other factors are not included in the analysis (for example, cardiovascular risk factors and family history of CVD), which may distort the relationship between BMI and cardiovascular morbidity and mortality [35].

This study has several limitations. Firstly, the cross-sectional study design did not allow us to study the causal links between obesity and other cardiovascular risk factors. Moreover, for practical feasibility, only patients in a narrow age range were invited to participate in the LitHiR prevention program. This limited the data collection regarding the association between obesity and cardiovascular risk factors to the middle-aged population. Lastly, the results of this study cannot be generalized to other populations.

## 5. Conclusions

The most prevalent cardiovascular risk factor among middle-aged Lithuanian men was dyslipidaemia, with a surprisingly high prevalence in all BMI and WC groups. All risk factors except for smoking (arterial hypertension, dyslipidaemia, diabetes mellitus, and metabolic syndrome) were significantly more common in men with obesity or overweight compared to those with normal weight, as measured by BMI. A similar trend was observed in WC groups, where these risk factors were more prevalent in men with increased WCs or abdominal obesity than those with normal WCs. Smoking was the only risk factor most prevalent in subjects with low or normal weight according to BMI. It was also more prevalent in the normal WC group compared to the increased WC or abdominal-obesity groups, but ABSI values associated with higher mortality were more prevalent among smokers than non-smokers.

## Figures and Tables

**Figure 1 medicina-58-01718-f001:**
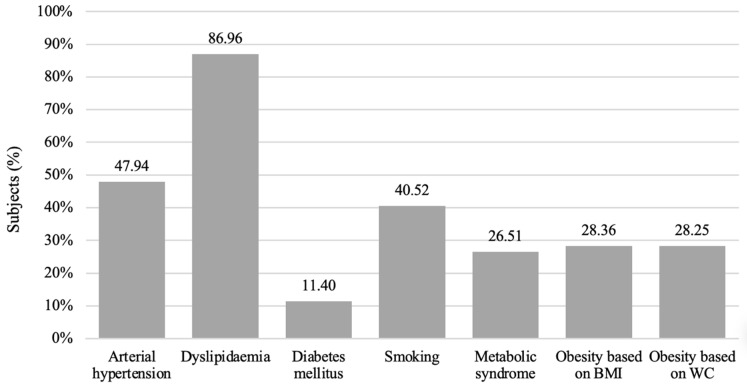
Prevalences of cardiovascular risk factors.

**Figure 2 medicina-58-01718-f002:**
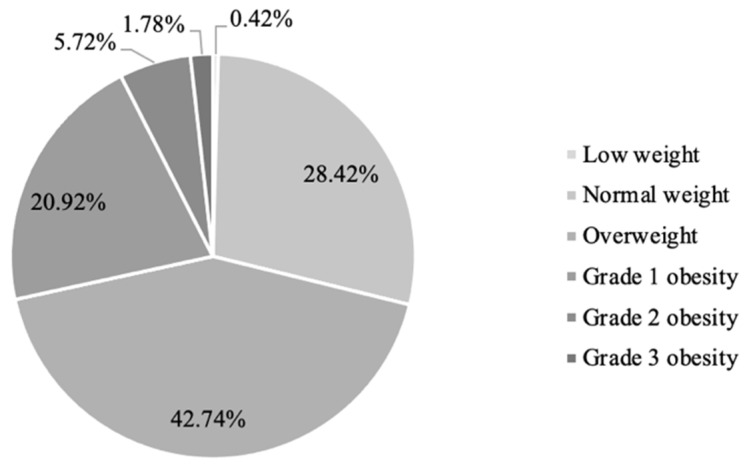
BMI distribution.

**Figure 3 medicina-58-01718-f003:**
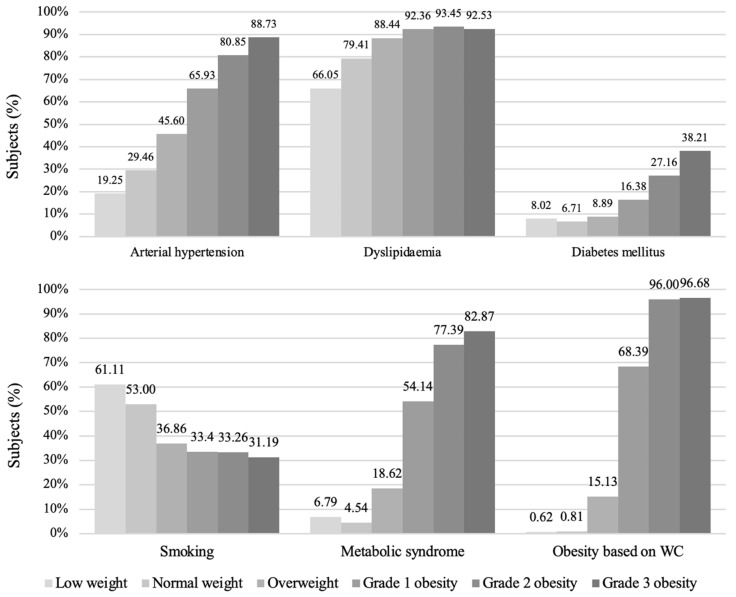
Prevalences of cardiovascular risk factors in BMI groups.

**Figure 4 medicina-58-01718-f004:**
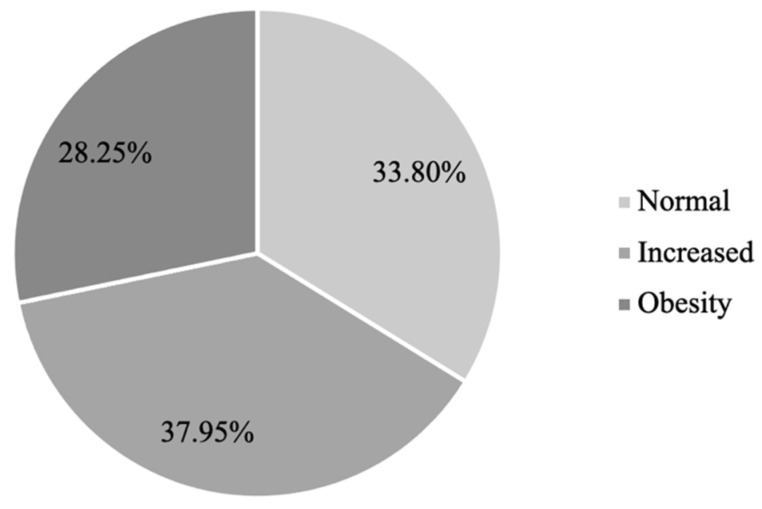
WC distribution in the studied population.

**Figure 5 medicina-58-01718-f005:**
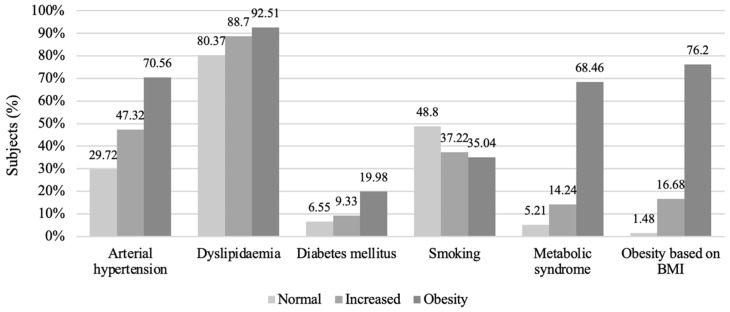
Prevalences of cardiovascular risk factors in the WC groups.

**Figure 6 medicina-58-01718-f006:**
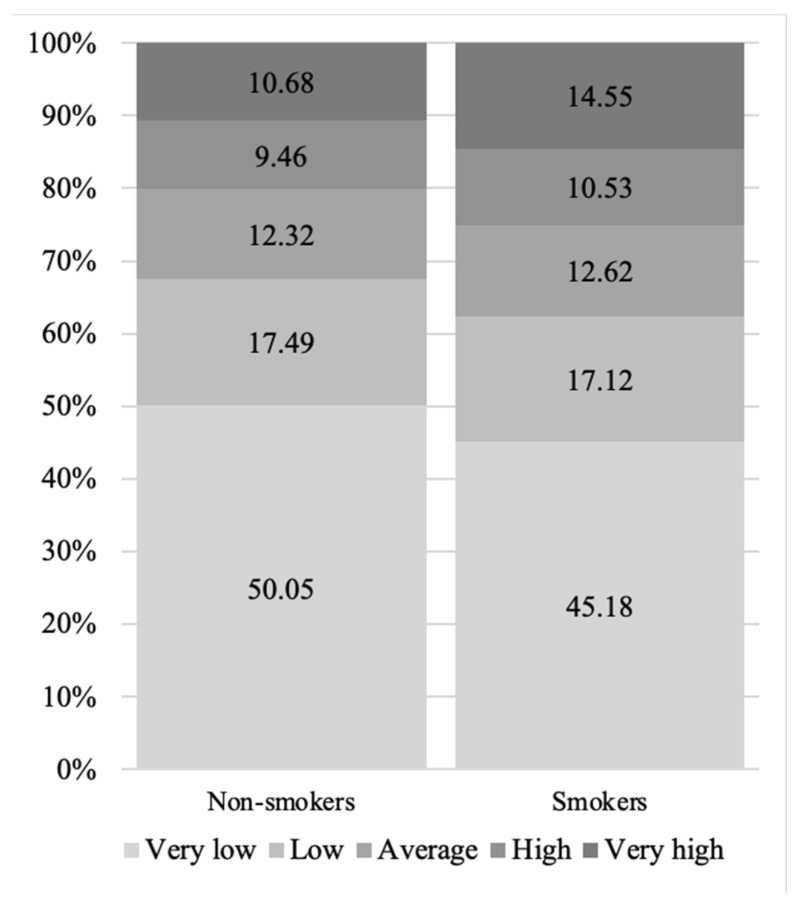
Mortality risk according to ABSI Z-scores for non-smokers and smokers.

**Table 1 medicina-58-01718-t001:** General characteristics of the study sample.

	Sample (*n* = 38,412)
	Mean ± SD
WC (cm)	96.76 ± 12.42
BMI (kg/m^2^)	27.90 ± 4.72
sBP (mmHg)	132.80 ± 15.58
dBP (mmHg)	83.25 ± 9.62
HR (bpm)	71.71 ± 9.06
Fasting glycemia (mmol/L)	5.56 ± 1.26
Total cholesterol (mmol/L)	5.84 ± 1.20
LDL-C(mmol/L)	3.73 ± 1.06
HDL-C (mmol/L)	1.40 ± 0.46
Triglycerides (mmol/L)	1.74 ± 1.40
SCORE	1.90 ± 1.74

Abbreviations: BMI—body mass index; WC—waist circumference; SBP—systolic blood pressure; DBP—diastolic blood pressure; HR—heart rate; LDL-C—low-density lipoprotein cholesterol; HDL-C—high-density lipoprotein cholesterol; SCORE—Systemic Coronary Risk Evaluation index.

**Table 2 medicina-58-01718-t002:** General characteristics based on BMI.

	LW (*n* = 325)	NW (*n* = 13,659)	OW (*n* = 19,001)	G1O (*n* = 13,005)	G2O (*n* = 5529)	G3O (*n* = 2442)	NW vs. OW	NW vs. G3O	OW vs. G3O	NW vs. LW	LW vs. G3O
	Mean ± SD	Mean ± SD	Mean ± SD	Mean ± SD	Mean ± SD	Mean ± SD	*p*	*p*	*p*	*p*	*p*
Age (years)	47.60 ± 4.68	46.82 ± 4,42	48.98 ± 4.38	47.06 ± 4.39	47.11 ± 4.34	46.95 ± 4.35	**	**	**	**	**
WC (cm)	76.55 ± 8.69	85.81 ± 7.42	95.57 ± 7.28	106.11 ± 7.45	117.09 ± 8.41	129.45 ± 12.12	**	**	**	**	**
sBP (mmHg)	122.43 ± 15.20	127.68 ± 13.68	131.89 ± 14.45	137.74 ± 15.95	142.87 ± 16.93	148.59 ± 18.37	**	**	**	**	**
dBP (mmHg)	77.47 ± 8.99	80.28 ± 8.53	82.85 ± 9.05	86.09 ± 9.96	88.65 ± 10.64	90.99 ± 11.28	**	**	**	*	**
HR (bpm)	76.00 ± 12.44	71.29 ± 9.19	71.22 ± 8.80	72.39 ± 9.03	73.36 ± 9.11	75.97 ± 10.02	0.982	**	**	**	0.982
Fasting glycemia (mmol/L)	5.49 ± 1.26	5.36 ± 1.07	5.47 ± 1.07	5.77 ± 1.41	6.20 ± 1.93	6.54 ± 2.19	**	**	**	0.763	**
TC (mmol/L)	5.08 ± 1.35	5.63 ± 1.16	5.90 ± 1.19	6.02 ± 1.22	5.89 ± 1.16	5.82 ± 1.17	**	*	0.499	**	**
LDL-C (mmol/L)	3.03 ± 1.12	3.51 ± 1.05	3.81 ± 1.06	3.87 ± 1.05	3.73 ± 1.01	3.71 ± 1.01	**	**	0.198	**	**
HDL-C (mmol/L)	1.61 ± 0.63	1.59 ± 0.52	1.39 ± 0.43	1.26 ± 0.36	1.18 ± 0.32	1.15 ± 0.30	**	**	**	0.999	**
TGs (mmol/L)	1.12 ± 0.56	1.27 ± 0.90	1.70 ± 1.26	2.19 ± 1.77	2.47 ± 1.86	2.40 ± 1.49	**	**	**	*	**
SCORE	1.67 ± 1.5	1.73 ± 1.57	1.83 ± 1.65	2.12 ± 1.91	2.26 ± 2.07	2.63 ± 2.35	**	**	**	0.996	**

* Level of statistical significance: <0.05. ** Level of statistical significance: <0.001 Abbreviations: BMI—body mass index; WC—waist circumference; sBP—systolic blood pressure; dBP—diastolic blood pressure; HR—heart rate; TC—total cholesterol; LDL-C—low-density lipoprotein cholesterol; HDL-C—high-density lipoprotein cholesterol; TGs—triglycerides; SCORE—Systemic Coronary Risk Evaluation index; LW—low weight; NW—normal weight; OW—overweight; G1O—grade 1 obesity; G2O—grade 2 obesity; G3O—grade 3 obesity.

**Table 3 medicina-58-01718-t003:** General characteristics in the WC groups.

	Normal WC (*n* = 12,498)	Increased WC (*n* = 11,908)	Obesity (*n* = 29,555)	1 vs. 2 vs. 3	1 vs. 2	1 vs. 3	2 vs. 3
	Mean ± SD	Mean ± SD	Mean ± SD	*p*	*p*	*p*	*p*
Age (yrs.)	46.66 ± 4.41	47.00 ± 4.38	47.26 ± 4.36	**	**	**	**
BMI (kg/m^2^)	24.06 ± 2.60	27.54 ± 2.63	32.96 ± 4.24	**	**	**	**
sBP (mmHg)	127.72 ± 13.59	132.24 ± 14.56	139.65 ± 16.58	**	**	**	**
dBP (mmHg)	80.29 ± 8.44	83.02 ± 9.07	87.09 ± 10.34	**	**	**	**
HR (bpm)	71.03 ± 8.96	71.41 ± 8.94	72.93 ± 9.23	**	**	**	**
Fasting glycemia (mmol/L)	5.35 ± 1.03	5.50 ± 1.08	5.90 ± 1.63	**	**	**	**
TC (mmol/L)	5.65 ± 1.16	5.91 ± 1.19	5.98 ± 1.22	**	**	**	**
LDL-C (mmol/L)	3.55 ± 1.05	3.81 ± 1.06	3.83 ± 1.04	**	**	**	0.281
HDL-C (mmol/L)	1.55 ± 0.51	1.39 ± 0.44	1.24 ± 0.35	**	**	**	**
TGs (mmol/L)	1.32 ± 0.97	1.73 ± 1.32	2.24 ± 1.73	**	**	**	**
SCORE	1.65 ± 1.50	1.85 ± 1.64	2.23 ± 2.04	**	**	**	**

** Level of statistical significance: <0.001. Abbreviations: BMI—body mass index; WC—waist circumference; sBP—systolic blood pressure; dBP—diastolic blood pressure; HR—heart rate; TC—total cholesterol; LDL-C—low-density lipoprotein cholesterol; HDL-C—high-density lipoprotein cholesterol; TGs—triglycerides; SCORE—Systemic Coronary Risk Evaluation index.

## Data Availability

The data presented in this study are available on request from the corresponding authors.

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
