# Peer review of "Prevalence of Cardiovascular Risk Factors in Middle-Aged Lithuanian Men Based on Body Mass Index and Waist Circumference Group Results from the 2006–2016 Lithuanian High Cardiovascular Risk Prevention Program"

_medicina, 2022, doi:10.3390/medicina58121718_

Round 1
Reviewer 1 Report
This review provides an analysis of the prevalence of cardiovascular risk factors (arterial hypertension, dyslipidaemia, diabetes, smoking, and metabolic syndrome) in middle-aged Lithuanian men and its differences in groups stratified by body mass index and waist circumference. The manuscript is interesting; however, the following concerns should be addressed:
The novelty of the manuscript is rather limited and should be highlighted.
- Analyses on women should also be done, added in this manuscript and compared both.
- Criteria for considering metabolic syndrome should be added.
- How was data normality tested?
- Are Lithuanian men included in the study following guidelines for dyslipidemia management? Please specify.
- What was the rationale for selecting this age range?
- Have genetic dyslipidemias such as familial hypercholesterolemia taken into account?
- Please consider the analysis based on the clustering of risk factors / comorbidities.
- Authors should emphasize the impact of their findings on the cardiovascular risk assessment and the global management of dyslipidemia.
Minor comments:
- Page numbers are missing.
- Tables layout should be improved.
Author Response
Dear reviewer,
First of all, we would like to thank you for your marks and comments about our manuscript. We really appreciate your help with improving our work. Without further do, here is our reply to the points that were made:
- On analyses of women: The Lithuanian High Cardiovascular Risk prevention program includes men from 40 to 54 years old and women aged from 50 to 64 years old. The reason behind such different age ranges for men and women is the typically later onset of cardiovascular risk and diseases in women. This results in the program being more cost-effective. It also corresponds to 2021 ESC guidelines on cardiovascular risk prevention in clinical practice. However, the difference in age between men and women limits our ability as researchers to properly compare both sexes in terms of cardiovascular risk factor prevalence, which is why we left out the female population in this study.
- Specific criteria for considering metabolic syndrome will be added to our manuscript.
- Data normality testing will be specified in our manuscript.
- Lithuanian men are included in this study following the guidelines on cardiovascular risk prevention – all men who fall in the program’s age range and who do not have overt cardiovascular disease are eligible to participate.
- We have not taken into account genetic dyslipidaemias. We will specify this in our manuscript.
- On importance of this manuscript: Cardiovascular disease is the leading cause of death in Europe and especially in previous soviet bloc countries such as Bulgaria, Romania, Estonia, Latvia, and Lithuania, which all are considered as very high-risk regions according to ESC guidelines. In 2019, in Lithuania, more than half of deaths (55.1%) were attributed to cardiovascular disease, which makes it the second largest proportion of cardiovascular deaths in European Union. As this is such an important issue, naturally many studies have been made that explored the relationship between different cardiovascular risk factors, and cardiovascular disease in Lithuanian population to better understand the risk profile in this country. To our knowledge, our study is the first one that explores obesity and its relation to the other main causal CVD risk factors in a large sample of middle-aged Lithuanian men. We also considered our results relevant to the readers of this journal as it is the official journal of Lithuanian University of Health Sciences and could provide rationale for management of cardiovascular risk factors and especially obesity in the region.
- We corrected missing page numbers and improved table design.
Kind regards,
Authors of the manuscript
Reviewer 2 Report
B”SD
Prevalence of cardiovascular risk factors in middle-aged Lithuanian men in body mass index and waist circumference groups. Results from 2006-2016 Lithuanian High Cardiovascular Risk prevention program
Egidija Rinkūnienė1*, Emilija Petrulionytė1, Vilma Dženkevičiūtė1, Žaneta Petrulionienė1, Augustė Senulytė1, Roma Puronaitė1, Aleksandras Laucevičius1
1 Faculty of Medicine, Vilnius University, Vilnius, Lithuania
Initial Review 10/24/2022
First let me express appreciation to the Editor for the opportunity to review the manuscript which is drawn from a cohort of middle age men assessed upon entry into a risk prevention program over the decade ending in 2016. Subjects were screened by cutoff thresholds mostly for components of the metabolic syndrome and for smoking. The prevalence of these findings was then analyzed by BMI and waist circumference (WC). Their findings are quite as expected from similar cohorts worldwide, except that: “Smoking was the only risk factor most prevalent in subjects with low or normal weight according to BMI and normal WC.”
However, it seems to me that the association of smoking with trimmer WC could be equally explained by the well known association of BMI with smoking and:
1. the use of categories for BMI, which is a continuous measure.
2. the high correlation of BMI and WC (0.7-0.9)
To better understand the anthropometric associations of smoking, ABSI, an index that adjusts WC for BMI has been available and extensively validated. (1,2) ABSI has also been studied in the context of the MS. (3-5) To my knowledge there has been no study of associations with smoking, which provides the authors the opportunity for novel findings.
Once the WC and smoking paradox is resolved, I would be glad to further review this interesting paper.
Selected References:
1. Krakauer NY, Krakauer JC. A new body shape index predicts mortality hazard independently of body mass index. PLoS One. 2012;7(7):e39504. doi: 10.1371/journal.pone.0039504. Epub 2012 Jul 18. PMID: 22815707; PMCID: PMC3399847.
Annotation: A body shape index (ABSI) is a power law based on weight and height that normalizes waist circumference (WC) for weight and height. In a large USA population sample, mortality increased significantly with ABSI and independently from BMI. Since publication, over 360,000 views to this article and 460 citations have been reported at the journal web site.
2. Christakoudi S, Tsilidis KK, Muller DC, et al. A Body Shape Index (ABSI) achieves better mortality risk stratification than alternative indices of abdominal obesity: results from a large European cohort. Sci Rep. 2020;10(1):14541. Published 2020 Sep 3. doi:10.1038/s41598-020-71302-52.
3. Krakauer NY, Krakauer JC. Anthropometrics, Metabolic Syndrome, and Mortality Hazard (2018). Journal of Obesity. 2018;2018:9241904. doi:10.1155/2018/9241904..
4. Bertoli S, Leone A, Krakauer NY, Bedogni G, Vanzulli A, Redaelli VI, et al. (2017) Association of Body Shape Index (ABSI) with cardio-metabolic risk factors: A cross-sectional study of 6081 Caucasian adults. PLoS ONE 12(9): e0185013.
5. Nagayama D, Sugiura T, Choi SY, Shirai K. Various Obesity Indices and Arterial Function Evaluated with CAVI - Is Waist Circumference Adequate to Define Metabolic Syndrome? Vasc Health Risk Manag. 2022 Sep 12;18:721-733. doi: 10.2147/VHRM.S378288. PMID: 36120718; PMCID: PMC9480599.
Author Response
Dear Reviewer,
Thank you for your comments on this study. We calculated ABSI and ABSI z-score among men who smoke and non-smokers. We found a higher mortality risk (based on ABSI z-score) in smokers compared to non-smokers (p<0.001), which adds an explanation to the smoking-obesity paradox. We updated our manuscript with this finding.
Kind regards,
Authors of the publication
Round 2
Reviewer 1 Report
Authors have satisfactorily addressed the vast majority of my comments.
Author Response
Dear reviewer,
Thank you for your review.
Kind regards,
Authors of publication
Reviewer 2 Report
B"sd
please see file

Author Response
Dear reviewer,
Thank you for your detailed review and apt remarks. We improved our manuscript according to the corrections made. Unfortunately, due to time constraints set by the journal, we were not able to supplement our paper with recommended analysis. ABSI has piqued our interest and we plan to make a separate analysis with the cohort of the prevention program. We thank you for providing us with the information and inspiration to further explore this interesting topic.
Sincerely,
Authors of the publication